# Changes in parenting behavior in the time of COVID—19: A mixed method approach

**Luiza Mesesan-Schmitz**[1], **Claudiu Coman**[1]*, **Carmen Stanciu**[2], **Venera Bucur**[2], **Laurentiu Gabriel Tiru**[2], **Maria Cristina Bularca**[1]

1 Faculty of Sociology and Communication, Transilvania University of Brasov, Brasov, Romania, 2 Faculty of Sociology and Psychology, West University of Timisoara, Timisoara, Romania

* claudiu.coman@unitbv.ro

**Data Availability Statement:** All relevant data are within the manuscript and its Supporting information files.

## Abstract

This study was designed to explore mothers' perceptions about changes in parenting behavior in the middle of the pandemic COVID 19 period. Based on the convergent mixed-method design and Parental Stress model, we illustrated these changes by taking into account the impact of the pandemic perceived by mothers and the resources they had available. Research on parenting changes was important in the Romanian context because, in that challenging period, there were no regulations to safeguard parents, especially single parents as mothers. Mothers experienced increased levels of stress, some of them having to leave their jobs to stay at home with their children. Other mothers needed to work from home and in the meantime to take care of their children. In this context we wanted to illustrate the possible changes that occurred in their parenting behavior during the pandemic period. Results from the quantitative survey showed that there is a moderate correlation between the negative impact felt by mothers and the negative changes in their parenting behavior, and this correlation was diminished by a series of resources such as: social support, parenting alliance, or high income. Qualitative data provided better understanding of mothers' parenting behavior by showing that mothers shared both positive and negative experiences during the pandemic, regardless of the general trend mentioned. As shown by the quantitative data, the qualitative data also showed that mothers who felt more strongly the impact of the pandemic reported more negative changes in their parenting behavior. The positive changes most frequently stated involved expressing affection and communicating more often on various topics, carrying out leisure activities or activities meant to help with the personal development of the child, and involving children in domestic activities. Mothers mostly described negative aspects such as too much involvement in school life, increased control and surveillance of children, especially when it comes to school related activities and to the time children were allowed to spend on their digital devices. These changes led to conflicts and sometimes, mothers resorted to discipline practices. In addition to the resources identified in quantitative research, mothers with higher education and medium–high income also turned to specialized resources (psychologists, online courses, support groups) in order to manage conflicts, them being able to see the challenges of the pandemic as an opportunity to develop and improve the relationship with their children.

**Funding:** The authors received no specific funding for this work.

**Competing interests:** The authors have declared that no competing interests exist.

## Introduction

The COVID– 19 pandemic has generated changes in almost every life sector, one of the most affected being the family sector. In this regard, the routines of families were disrupted by the pandemic and the measures taken in order to overcome it. Taking into account these aspects, the purpose of our paper was to identify the changes in the parenting behavior of Romanian mothers whose children are under the age of 18 in the context of the pandemic. In other words, our paper focused on examining if during the COVID– 19 pandemic, there have been changes in the parenting behavior of Romanian mothers, and on analyzing which aspects of their parenting behavior changed during this period. In this regard, while considering the concept of parenting behavior, we also focused our study on disciplinary practices applied by mothers to children, which referred to: ignoring the child due to misbehavior, taking away the child's privileges, sending the child to his/her room as punishment, screaming at the child when he/she has done something wrong, calmly explaining why the child's behavior was wrong, giving the child time-out, giving the child extra chores as punishment.

Thus, due to schools' closure, children had to spend all their time at home with their parents, who either lost their jobs or started to work from home [1]. In the middle phase of the pandemic, in Romania, some schools opened again and some did not. Thus, in the context of school or return to work policies, according to Law nr.55 from 2020 of the Romanian Parliament for preventing and combating the effects of the COVID-19 [2], within educational institutions the measures which were imposed were general, they depended on the evolution of the pandemic and on each educational institution. One of the measures stated that: subject to the analysis of the epidemiological situation at national level carried out by the Ministry of Health and based on the decision of the National Committee for Emergency Situations, by order of the Minister of Education, the suspension of activities that require the physical presence of pre-school children, preschoolers and students in educational units can be ordered, and the didactic activities will continue in the online system. Other measures stated that: during the state of alert, which was still available in 2021, and until the restrictions on public gatherings are removed by the relevant authorities, the pre-university education units organize activities from the education plans in the online environment, that in order to ensure equal access to education, school inspectorates and pre-university education units have the obligation to ensure educational resources for students who do not have access to technology, in accordance with the instructions of the Minister of Education and Research, and that the national exams of students, which involved face to face interaction, had to be taken face to face but under health protection conditions [2].

In this regard, when it comes to parenting, there were some uncertainties created either by the possibility to develop online the educational process in schools where this process was carried out face to face, or by the pressure mothers felt because their children had different school schedules or they were enrolled in schools which resorted to different ways of teaching. In this regard, as previous studies conducted before the pandemic mentioned, teachers must not resume their teaching methods to only one facilitation style, but instead they should adapt the methods according to the needs of the children [3], and in order to help students develop educational values, teachers should promote cooperation, mutual support and responsibility [4].

Furthermore, the pandemic changed the daily routines of mothers: they had to work from home, exclusively online, or in a hybrid system, they had to follow new rules and fulfil new tasks, they had to fulfil other roles in relation with their children, such as the role of the teacher, and among all these, there was also the pressure exerted by the social environment in terms of following health rules, or in terms of the possibility to contract the virus. Hence, parents in general and mothers in particular, faced the challenge of finding the best solutions in order to

keep an eye on their children, and to spend quality time with them, especially in the cases in which mothers started to work face to face again or in a hybrid system. Thus, considering that previous studies showed that in general, the difficulties encountered by parents in managing the time spent with children can negatively influence the social development of the children [5], an analysis of the way mothers spent time and interacted with their children in the context of the pandemic becomes very important.

Hence, the way mothers behaved in relation with their children might have also been influenced by the access they had to several resources. Considering the restrictive measures implemented by governments, and the results of a previous study which showed that people who live far from the center of the city have access to a wider range of public services than people who live near the city center [6], the area in which families lived in the time of the pandemic might have had a role in influencing the daily routines of families.

In this context, the challenges and high levels of stress that parents had to face during the pandemic can influence their relationships with their children and can create disruptions in the way parents and children relate [7]. Furthermore, certain aspects such as the age of the children or the jobs of the mothers, can lead to inequalities between mothers and fathers regarding the time spent on childcare [8], previous studies revealing that during the pandemic, women performed most of the household duties [9], and that, even though both parents had to be more involved in the lives of their children, mothers continued to have greater involvement in the process of childcare provision than fathers [10].

Considering the parenting behavior in the Romanian context during the pandemic, very few studies were conducted on this subject. A previous study conducted on parents and children from the epicenter of the COVID -19 outbreak in Romania- Suceava [11], showed that, compared to the period prior to the pandemic, parents and children focused more on pending time in free time activities, chores, or social interactions, that they encountered difficulties in maintaining social dynamics, managing emotions, dealing with school issues, or motivating children to comply with parental norms. Another study, which focused on the feelings and personality traits of parents of primary school pupils [12], revealed that, during the pandemic, as the parents' level of anxiety increased, their anger level also increased, but their level of self-efficacy decreased. Another study, which focused on the parent–child relationship and their use of technology and media [13], discovered a favorable association between a permissive parenting style and parents' use of technology, and highlighted the fact that children were more likely to utilize technology if their parents used it frequently, and that a democratic parenting approach could have a positive impact on children's use of technology.

Taking into account these aspects, the purpose of our paper was to identify the changes in the parenting behavior of Romanian mothers whose children are under the age of 18 in the context of the pandemic. In other wordson the changes that occurred in the parenting behavior of mother from Romania during the pandemic. The data was collected online, the questionnaires being shared on certain private Facebook groups, dedicated to mothers. Thus, the method of using social media for collecting data has some weaknesses, as the sample can't be representative for the population who is targeted and the people who are part of some social media groups represent some particular type of people, especially those who are more socially involved. Even more, the private Facebook groups are moderated and members must adhere to the rules imposed by the moderators, which could influence the topics discussed by members and the type of people who would consider joining those groups. But in the pandemic period, it was difficult to gather responses from the mothers in another way. However, by gathering data through Facebook groups, the authors of this study were able to get in touch with the community of mothers from Romania, and also to gather some insights about the supportive mechanism that exist within their community.

## Theoretical framework

In order to address the subject of parenting behavior, our theoretical framework revolves around the Parental Stress Model developed by Richard Abidin [14]. According to the model, parenting behavior is influenced by multiple variables. While recognizing the influence of environmental, behavioral or developmental variables, the author also mentions that the parenting behavior is mediated by a series of resources.

Hence, by analyzing the proposed model, we can observe that elements such as work, life events, marital relationships, daily struggles, or the environment have an influence on the parenting role. The parenting role variable refers to the beliefs and expectations that parents have with regard to how they should act as parents, how they should behave and interact with their children. Thus, while bearing in mind a certain image of how they should parent, parents analyze the harm and the benefits associated with their behavior, and their self—assessment process further determines the stress level felt by them [14].

Taking into account these aspects, in the parenting behavior model, the stress felt by parents is seen as an element that motivates them to make use of the resources they have to develop their parenting behavior. Thus, the resources dimension of the model includes social support, parenting alliance, parenting skills, material resources, and cognitive coping [14]. Thus, within the paper we focused on these specific resources because we believed that by studying them we could understand better the factors that influence parenting behavior, and because these type of resources, if managed correctly, could contribute to generating positive parenting behaviors.

## The concept of parenting behavior

Over time, the subject of parenting was approached from varied perspectives which often refer to parenting styles and parenting dimensions. In the context of parenting, the concept of parenting styles is defined, broadly, as a "constellation of attitudes toward the child that are communicated to the child and that, taken together, create an emotional climate in which the parent's behaviors are expressed" [15]. Thus, such behaviors do not refer only to the actions that parents intentionally take in order to fulfil their duties as parents, but they also refer to the gestures they make, their emotional responses or the changes in their tone of voice.

Due to its complexity, the concept of parenting was also analyzed from the perspective of the dimensions it encompasses. Scholars defined and identified many dimensions of the concept [16], but there is a high consensus on two main dimensions: parental support and parental control [17, 18]. The parental support dimension refers to various aspects such as bonding, warmth/caring, closeness, involvement, communication, autonomy [18–20]. Parental control is divided into psychological control- which refers to processes that involve emotions, attachment elements and the way the child thinks, and behavioral control- which refers to the behavior adopted by parents in order to control the behavior of their children [21]. Therefore, parental control support has sub—dimensions such as monitoring, overprotection, discipline practices, harsh parenting [18, 20, 22]. Taking into account the aspects previously mentioned, the parenting behavior can be characterized from a positive and negative perspective depending on its impact on children. For example, certain practices of parents such as being involved in their children lives, setting clear rules, encouraging their children to develop autonomous behavior, their acceptance of the way children behave, cognitive competence can be included in the positive parenting category. On the other hand, excessive control, not paying attention to the children and neglecting them and also punishments, attitudes of rejection and certain expectations regarding the accomplishments of the children can be included in the negative parenting category [17, 23]. Moreover, the actions which fall into the category of parental support can be considered positive, and by their faulty manifestation or lack of manifesting them,

these actions become negative and can negatively impact children. For example, the lack of attention to the children can affect in the short and long term the psychological and physical health of children [24].

The positive behavior of parents is also found in the literature under the concept of positive parenting. In this regard, positive parenting is defined as the a continuous relation between the parent and the child which includes "caring, teaching, leading, communicating and providing for the needs of the child in a consistent and unconditional manner" [25]. In the context of negative parenting, relevant is the concept of dysfunctional or poor parenting. Hence, poor parenting refers to the behavior in which parents are either too permissive or too authoritarian, they are strict and firm in their decisions, and they can create unrealistic expectations when it comes to the behavior of their children [26]. Moreover, the concept of disciplinary practices can also fall within the category of negative behavior, due to the fact that such practices can refer to time-out, taking away certain privileges that children have, slapping them or yelling at them [27].

Considering the discipline practices, they are methods through which parents try to control the children's negative behavior and promote their positive behavior. In this regard, these practices could be violent or non–violent [28] and some of the discipline practices include: teaching children about good and bad behavior, making the child apologize, giving children time–out, taking away their privileges, using corporal punishment, expressing disappointment, shaming, yelling, withdrawing love for misbehavior, threatening them or promising a treat or a privilege [29]. Given the impact of such practices on children, a previous study conducted on mothers and children [29], found that practices such as corporal punishment, expressing disappointment and yelling were associated with more child aggression, and also that corporal punishment, expressing disappointment and shaming were associated with more child anxiety. Another study [28], conducted by UNICEF on mothers and children from low and middle–income countries, revealed that non-violent practices, mainly explaining why a behavior is wrong, were generally the most common practices used by mothers or caregivers, and only one in four mothers/caregivers believed that physical punishment (such as shaking the child, slapping the child) is needed in order to educate and manage the behavior of their children. Thus, taking into account the two categories of disciplinary practices, in our research we paid attention to the non-violent disciplinary practices, which include acts such as taking away privileges or explaining why something is wrong. Alt these kind of acts can have a positive impact on the outcomes of the children, and we supposed that during the pandemic, any changes in the frequency of these acts can increase frustration and negative emotions of the children and can have a negative impact on them. Details about the disciplinary practices that were measured in our study can be found in S1 Appendix.

Furthermore, parental involvement in school was also considered relevant in our study. Usually, the parental involvement in the education of children has a positive impact on the outcomes of children [30], even if the effect is largely dependent on the quality of this involvement [31–34]. Moreover, a previous study also found that, parental involvement predicted the academic success and mental health of adolescents through behavioral and emotional engagement [35]. In this regard, in our study we wanted to see if during the pandemic there was an increase in the involvement of mothers in school activities as a strategy to help children to deal with the new situation and to have a good mental health.

## Materials and methods

### Purpose, research questions and hypotheses

Considering the Parental Stress Model, the changes from the external environment which took place during the pandemic could have increased the parents' stress level and could have

influenced their parenting behavior. According to the assumptions previously mentioned, the purpose of our research was to identify if in middle phase of the COVID—19 pandemic there have been changes in the Parenting Behavior (PB) of mothers from Romania whose children are under the age of 18, and what aspects of their parenting behavior changed in the time of the pandemic. In order to conduct the research, we used the convergent mixed methods design, parallel-databases variant [36]. This variant involves quantitative and qualitative data collected simultaneously to obtain a better understanding of the same topic. We collected quantitative data in order to assess if the way mothers were impacted by the COVID– 19 pandemic (PI), determined changes in their Parenting Behavior (PB), and to identify the elements that were used as resources to diminish this impact. Hence, the research questions of the quantitative survey and the hypotheses related to them were:

**RQ1. What is the relationship between the impact of the pandemic perceived by mothers and their subjective perception about the changes in their parenting behavior?**
*(H₁) Mothers who perceive a higher impact of COVID-19 (PI) will report higher scores related to negative changes in parenting behavior (PB)*
*(H₂) Mothers who perceive a higher impact of COVID-19 (PI) will report higher scores related to Discipline Practices (DP)*
*(H₃) Mothers who perceive a higher impact of COVID-19 (PI) will report lower scores related to Parent Involvement in School (PIS)*

**RQ2. What are the types of resources which managed to diminish the relationship between the impact of the pandemic as perceived by mothers and their subjective perception about the changes in their parenting behavior?**
*(H₄) Mothers who have access to a series of resources such as social support, parenting alliance, parenting skills, material resources, cognitive coping, will report lower scores related to the intensity of the relation between the impact of the pandemic as perceived by mothers and their subjective perception about the changes in their parenting behavior/discipline practices*

Considering the fourth hypothesis of our research, a moderating relationship was introduced due to the fact that the parenting behavior could be influenced by many factors/ resources. In this context, the moderating relationship was introduced in order to obtain a better view of the complexity of relationship between the impact of the pandemic and their perception about the changes in their parenting behavior/discipline practices. Furthermore, by looking at the resources mothers had during the pandemic, we were able to better understand the circumstances in which the relationship between the variables is more or less strong/ pronounced.

The qualitative data were collected in order to provide more details about the changes in parenting behavior and the resources used by mothers during the pandemic. In the context of the qualitative research, the next question was addressed:

**Q1. How did mothers experience changes in parenting behavior during the pandemic and what types of resources have they used?**

## Data collection method and sample

The population for the quantitative study was selected in a non-probabilistic way and was comprised of 276 mothers from Romania whose children are under the age of 18 years old. Data was collected online through varied Facebook groups addressed to mothers, in the period January–April 2021. The groups were private Facebook groups created for mothers from

Romania. Some of the authors of this paper were enrolled in those groups and they shared the questionnaire there. Within those groups, the mothers usually share their experiences as mothers as well as the difficulties they encounter in the relationship with their children." The period of recruitment started on 15/01/ 2021 and ended on 24.04. 2021. The research received the approval of the Ethics Commission in social research from Transilvania University of Brasov, Romania, nr.22 from 15.01.2021. The participants provided written informed consent.

The details about the instrument used, the questionnaire, are presented in S1 Appendix.

For qualitative data, an open-ended questionnaire was sent online on the same Facebook groups. The mothers who responded to the quantitative survey did not respond to the qualitative survey. Quantitative and qualitative data were analyzed separately using specific methods of data analysis. There are several ways to integrate data [36, 37], but in this study, the integration occurred during the interpretation stage, in the *Conclusions and discussions* section.

The majority of the participants to the quantitative survey were mothers from the urban area (78.3%), with tertiary education (79.4%), in the early and middle years of adulthood (50%), who have children who attend school (71%) with medium (46%) and high income (36.2%). Participants indicated that they belong to the nucleus family (70.7%), with one child (49.3%) or more children (50.7%). Thus, 77.5% of participants reported that at that time they were working. Even more, 45.8% of them worked in the hybrid system (at home and at work) and 32.2% exclusively at work. Detailed sample characteristics for the quantitative survey are provided in Table 1.

In the qualitative survey 24 mothers participated. The data gathered was coded by two of the authors of the paper.

The sample is highly educated, 23 participants have tertiary education and only one has a secondary education, and they are aged between 30–49 years. The socio-economic status of most participants is relatively affluent (22 of participants have medium-high income) and the majority of them live in cities (21 of participants). Moreover, 18 women had employee status and 6 of them were working in the hybrid system, while 5 of them worked exclusively at work. Next, 16 women had children attending school and 8 of them had younger children. A table comprising detailed socio-demographic information of the participants to the qualitative survey is listed in S2 Appendix.

## Measures

**Perception of the pandemic impact (PI).** The impact of the COVID– 19 pandemic as perceived by mothers was analyzed using the Coronavirus Impacts Questionnaire (CIQ), the shortened 6-item version [38]. The shortened version contains two items from each of the three factors (financial (F), resource (R), and psychological impact (P)). Each item is presented with options from 1–7 anchored by "1 = not true of me at all" and 7 = "very true of me." In the present study, Cronbach's alpha is 0.80. An average score was computed, where the higher score indicated a higher negative impact of the pandemic as perceived by mothers.

**Parenting behavior (PB).** We created an ad hoc index that assessed the perception of mothers about the parenting behavior changes during the pandemic period. An average index [39] was computed from 8 items which represent dimensions of the parenting behavior concept based on the literature review [19, 21] such as: the ability to be a good parent, the ability to offer children affection, to communicate with them, to care for them, to monitor and control/supervise them, the ability to solve conflicts, to engage in fun activities with their children, and to engage in activities which involve listening and understanding of the children's needs. The scale was built based on the semantic differential technique [40], where each item represented a dimension of the concept on a 1 to 5 rating scale, in which 1 represents the agreement

**Table 1. Sociodemographic characteristics of the respondents to the quantitative survey (N = 276).**

|  | Category | Count | Percent |
|---|---|---|---|
| Age | 19–30 years old | 30 | 10.9% |
|  | 31–40 years old | 138 | 50% |
|  | 41–58 years old | 106 | 38.4% |
|  | n.r | 2 | .7 |
| Children attending school (over 6 years) | They have | 196 | 71% |
|  | They do not have | 80 | 29% |
| Family structure | Single mother | 18 | 6.5% |
|  | Nuclear family | 195 | 70.7% |
|  | Extended family or other combinations | 63 | 22.8% |
| Residence environment | Urban | 216 | 78.3% |
|  | Rural | 60 | 21.7% |
| Levels of education | Primary education | 4 | 1.4% |
|  | Vocational school | 8 | 2.9% |
|  | Secondary education | 45 | 16.3% |
|  | Tertiary education | 194 | 70.3% |
|  | PhD | 25 | 9.1% |
| Number of children | One child | 136 | 49.3% |
|  | More children | 140 | 50.7% |
| Income | Under 2000 Ron (low) | 29 | 10.5% |
|  | 2000–4000 Ron (medium) | 127 | 46% |
|  | Over 4000 Ron (high) | 100 | 36.2% |
|  | n.r. | 20 | 7.2% |
| Employment situation | Employed | 214 | 77.5% |
|  | Unemployed | 8 | 2.9% |
|  | Housewife | 54 | 19.6% |
| Way of working | Exclusively at home | 44 | 20.6% |
|  | Hybrid (at home and at work) | 98 | 45.8% |
|  | Exclusively at work | 69 | 32.2% |
|  | n.r. | 3 | 1.4% |

with the statement on the left and 5 the agreement with the statement on the right. Negative changes in PB were mentioned on the left, and positive changes in PB on the right. Thus, high values of the index variable indicate that mothers perceive positive behavioral changes, and low values of the index variable indicate that mothers perceive negative changes in parenting behavior. The validity of the construct was tested with Factorial Analysis (KMO = 0 .91, Method of Principal Components, Eigenvalues = 5.27, 65.92% of explained variance). Cronbach's alpha is 0.92.

**Discipline practices (DP).** Discipline practices were evaluated using an adapted form of Alabama Parenting Questionnaire (APQ), Other Discipline Practices-dimension Parent developed by Shelton, Frick and Wootton [22]. We used the scale items, but we rated them differently than the way they were rated on the original scale. We asked mothers about the changes related to the frequency of various discipline practices (7 items) compared to the time before the pandemic. The disciplinary practices to which we referred include: ignoring the child due to misbehavior, taking away the child's privileges, sending the child to his/her room as punishment, screaming at the child when he/she has done something wrong, calmly explaining why the child's behavior was wrong, giving the child time-out, giving the child extra chores as

punishment. Each item was rated on a 6 point Likert scale, where 0 (never, it is not the case), and 5 (much more often than before the pandemic). One item was recoded (5.You calmly explain to your child why his/ her behavior was wrong when he/she misbehaves). The validity of the construct was tested with Factorial Analysis (KMO = 0.82, Method of Principal Components, Eigenvalues = 3.19, 45.61% of explained variance). Cronbach's alpha is 0.79. An averaging score was computed, where the higher score indicated that the mothers used disciplinary practices more frequently than before, and a low score represented the fact that such practices were used less often. Respondents which rated the items with the score 0 (never, it is not the case) were no longer taken into consideration.

**Parent Involvement in School (PIS).**   Parent Involvement in School was assessed by using an adapted form of the Parent Involvement in School Scale, Parent Involvement with Child-dimension developed by Miller-Johnson et al. [22]. We used the items from the scale, but we rated them differently than the way they were rated on the original scale. We asked mothers about the changes related to the frequency of various items which reflect different ways to get involved in the school activity of the children (7 items) compared to the situation before the pandemic. Each item was rated on a 5 points Likert scale, where 1 means—less often than before the pandemic, and 5 means—more often. The validity of the construct was tested with Factorial Analysis (KMO = 0.87, Method of Principal Components, Eigenvalues = 4.87, 69.62% of explained variance). Cronbach's alpha is 0.79. An averaging score was computed, where the higher score indicated that during the pandemic, mothers engaged more in the school activities of their children than they used to before the outbreak of the pandemic, and a lower score shows that mothers engaged less in such activities.

**Resources.**   *Social support* was measured through one question: "In the pandemic period did someone help you with the domestic and childcare activities?" (1. Yes, 2. No). *Parenting alliance* was measured through one question. Besides the previously mentioned questions, we asked mothers to answer an open ended question, where mothers had to mention the persons who helped them. Thus, a dummy variable was built, with the code 1 –help from the husband or father of the child, and with the code 0 –other people.

For measuring *parenting skills competencies* we used an indirect measure, and we assumed that mothers who have at least a bachelor degree could also have certain parenting competencies. When considering this assumption, we took into account the results of previous studied which showed the influence of parental education on parents' behavior in relation with their children. In this regard, the results of a previous study revealed that a mothers' level of education influences not only the time spent with her children, but also the quality of that time and how she spends time with her children of different ages [41]. Furthermore, previous studies showed that the level of parents' education, as well as the parents' income, can have a positive influence on the behavior of the parents at home and on the affective relationship they have with their children [42], and that education, age, or income, can influence the way mothers' structure the home environment [43], and implicitly their parenting practices.

Therefore we used the variable related to the level of education with 1-primary school, and 2- at least bachelor degree. For measuring m*aterial resources* we recoded one question related to income with 1-low income (<2000 Ron), and 0- other levels of income. For measuring *cognitive coping* we used an indirect measure. We had two questions in the survey regarding the way mothers felt about themselves considering their role as mothers before and during the pandemic. In the case of mothers who used terms which expressed a better perception about themselves compared to the period before the pandemic, we considered that they reacted in a positive manner to the changes generated by the pandemic, and that they managed to develop coping strategies in order to deal with the situation. These mothers received the code 1, and the mothers who did not report these changes received the code 0.

The data were analyzed using IBM SPSS Statistics (version 23). For main measures as Perception of the pandemic impact (PI), Parenting Behavior (PB), Discipline Practices (DP), Parent Involvement in School (PIS) were built index variables as the mean of item's scale's scores. The items for each scale related to the concepts are presented in S1 Appendix. Where the scales were used in their initial form, fidelity was tested. Where the scales were adaptations of the original forms, validity was tested. To describe the values of the index variables, percentages, mean, and standard deviation were made. Furthermore, because our research is exploratory, at this stage we tried to identify possible links between variables with the Spearman correlation coefficient between the variables mentioned in the hypotheses, at the sample level, but also on the subsamples given by the categories of the resources variables. Later, they could be tested in a more complex explanatory model. We used this coefficient because it is less sensitive to the type of distribution of the variable scores (in the present case it can also be applied for variables that do not have a normal distribution).

For the *qualitative data*, several open ended questions were addressed regarding the elements with which their relationship with their children is associated, the aspects which improved or worsened in their relationship during the pandemic in general, and on each dimension of the parenting behavior in particular, and regarding the resources mothers resorted to in order to deal with the pandemic situation. The procedure used in the processing of the data was thematic analysis [44]. Using deductive coding, participants' responses were classified according to the dimensions of the parenting behavior, dimensions taken from the theoretical background of the quantitative survey (effective parenting, affection, communication, caring, monitoring and control, conflict, involvement, closeness). Taking into account these dimensions the responses were classified in two overarching thematic categories—perceived positive changes and perceived negative changes. Particular fragments from the open ended questions have been selected to illustrate positive or negative changes (S3 Appendix). For illustrating the resources to which mothers resorted to, inductive coding was used, participants' responses being classified into two categories: modern resources and traditional resources. Moreover, considering the qualitative analysis, we used inter-coder reliability. In this regard, the level of agreement was measured, and two authors who had the role of coders analyzed the qualitative codes established and connected them with the data gathered from the research.

Hence, in regard to the description of the qualitative data, there are two mentions we must make. Firstly, the process of integrating the answers into the dimensions of parenting behavior was difficult, due to the connections of these dimensions and their impact on each other, but we made this categorization based on the aspects the mothers emphasized in their answers. Secondly, the positive or negative changes were analyzed through the prism of the effects of the parenting behavior on children, even though some positive changes were not necessarily positive for mothers themselves or for their relationships with their partners.

## Results

### Quantitative data: Descriptive statistics

We calculated the means for the index variables (Table 2), and for their items (S1 Appendix), with the aim of identifying the behavioral changes at the level of the entire sample. Thus, the respondents reported an average *Perception of the pandemic impact* (PI) score of 2.69, indicating a low level of the negative impact as perceived by mothers. The average score for changes in *parenting behavior* (PB) is 3.76, indicating a high level of positive changes perceived by mothers. For all the items of this scale, we obtain scores above 3, them indicating positive

**Table 2. Descriptive statistics of the index variables.**

| Index variables | Mean | SD | Median | Min | Max |
|---|---|---|---|---|---|
| Perception of the pandemic impact (PI) | 2.69 | 1.43 | 2.33 | 1 | 7 |
| Parenting Behavior (PB) | 3.76 | .99 | 3.93 | 1 | 5 |
| Discipline Practices (DP) | 2.70 | .82 | 3 | 1 | 5 |
| Parent Involvement in School (PIS) | 3.41 | .62 | 3.14 | 1 | 5 |

changes. The lowest score was registered at the dimension: managing conflicts (3.46), it signals that mothers encountered some difficulties in this regard.

The sample average for *Discipline practice* (DP) is 2.08 indicating a lower level of perception about discipline practices compared to the situation before the pandemic. In other words, during the pandemic, the perception of mothers is that disciplinary practices were used less frequently than they were used prior to the pandemic. It is thus noted that mothers explained more frequently to their children, why *his/ her behavior was wrong when he/she misbehaves* (3.46).

The sample average for *Parent Involvement in School* (PIS) is 3.41, indicating a higher level of perceived involvement/engagement in the children's school activities during the pandemic. Hence, the research shows that mothers *talked with their children about doing their best at school* (3.51) and *talked with their children about their schoolwork* (3.47) more than before the pandemic. Mothers' experiences during the pandemic appeared to generally generate positive changes in parenting behavior, a high level of involvement in school life, and a lower level of discipline practices compared to the period previous to the pandemic.

However, even though there is this majority trend, our research also shows that there was a segment of mothers who reported negative changes in their lives: 19.9% mentioned that during the pandemic their parenting behavior changed in a negative manner (they have a score below 3), 17, 4% declared that the pandemic negatively influenced their life (they have a score above 4), 23,1% stated that the frequency of their disciplinary practices increased (score above 3), and 7,7% mentioned they engaged less in school activities than they used to before. Also, there is a segment of mothers who (approx. 20, 6%) said that they did not experience any behavioral changes in the relationship with their children (score between 3 and 3.5).

## Quantitative data: Bivariate correlations

### RQ1. What is the relationship between the impact of the pandemic perceived by mothers and their subjective perception about the changes in their parenting behavior?

Correlation analysis at general sample showed that mothers' negative perception of the pandemic impact (PI) were significantly positively related to mothers' negative perception of the changes in parenting behavior (PB) ($r = -.383$, $p = 0.01$) and mothers' perception of the increase of discipline practices (DP) during pandemic compared to the period before the pandemic ($r = .135$, $p = 0.027$). There is no correlation related to mothers' perception of the increase of involvement in school (PIS) ($r = -.004$, $p > 0.05$) (Table 3). In other words, mothers who perceived a higher impact of COVID-19 (PI) reported higher scores related to negative changes in parenting behavior (PB) and reported higher scores related to Discipline Practices (DP). Therefore the hypotheses $H_1$, $H_2$ were supported. Related to $H_3$, whether or not mothers perceived the pandemic to have a negative impact, they were involved more frequently in their children's school life. Therefore $H_3$ was not supported.

**Table 3. Bivariate correlations among index variables for the overall sample (n = 276) and sub-samples defined by resources.**

| Spearman coefficient | Resources | | Perception of the pandemic impact (PI) |
|---|---|---|---|
| Parenting Behavior (PB) | Total sample | | -.383** |
| | Social support | Yes | -.252** |
| | | No | -.491** |
| | Parenting alliance | Yes | -.122 |
| | | No | -.457** |
| | Levels of education | Primary education | -.639* |
| | | At least secondary education | -.371** |
| | Material resources (<2000 Ron) | Yes | -.561** |
| | | No | -.359** |
| | Cognitive coping | Yes | -.234 |
| | | No | -.373** |
| Discipline Practices (DP) | Total sample | | .135* |
| | Social support | Yes | .072 |
| | | No | .180* |
| | Parenting alliance | Yes | .120 |
| | | No | .143* |
| | Levels of education | Primary education | .608* |
| | | At least secondary education | .110 |
| | Material resources (<2000 Ron) | Yes | .361 |
| | | No | .110 |
| | Cognitive coping | Yes | .033 |
| | | No | .135* |
| Parent Involvement in School (PIS) | Total sample | | -.004 |

$**p < .01;$

$*p < .05$

***RQ2. What are the types of resources which managed to diminish the relationship between the impact of the pandemic as perceived by mothers and their subjective perception about the changes in their parenting behavior?***

Correlation analysis at sub-samples showed that the relationship between *perception of the pandemic impact* (PI) and mothers' perception of the changes in *parenting behavior* (PB) or *discipline practices* (DP) had a lower intensity among mothers who had *social support* in the pandemic period, who had *parenting alliance*, who had *at least secondary education level*, *material resources* greater than 2000 Ron (medium or high level) and had *cognitive coping* (Table 3). Therefore, it can be inferred that hypothesis $H_4$ was supported, in the case of mothers who had a series of resources during the pandemic, the impact of the pandemic on parenting behavior and discipline practices decreased in intensity.

## Qualitative data

### *Q1. How did mothers experience changes in parenting behavior during the pandemic and what types of resources have they used?*

Firstly, the qualitative data confirmed the results obtained through the quantitative research, mothers mentioning mostly positive changes related to the dimensions of parenting behavior. Generally, most mothers mentioned positive changes in their behavior (20 mothers). One

mother explained that, on the whole, her relationship with their children has deteriorated because of some negative changes in her parenting behavior, and that in this period she did not manage to be a "good mother" (single mother, with high-school studies without support from the father of the children). However, there were also 3 mothers who considered that the relationship with their children did not suffer significant changes, compared to the period before the pandemic. In this regard, we considered relevant the perception of one of the mothers: "*The relationship with my child was always special. I don't necessarily think that this period improved something. I could describe this period in terms of much more time spent physically with the child and teleworking. However, physical presence did not imply a more qualitative time spent with the child because the schedule at work was loaded and working hours increased.*"

Secondly, the data illustrates the dimensions where both positive and negative changes took place. Hence, mothers manifested affection towards their children more often, they engaged more in school activities but also in entertainment activities, they were more caring and attentive with their children, their become more close with their children because the communication between them improved, mothers tried to listen more and understand the needs of the children.

Mothers who mentioned positive changes in their parenting behavior had several reasons for doing so. Some mothers saw the pandemic period as an opportunity to improve their relationship with children and implicitly, to improve the way they behaved in relation to them. These mothers wanted to be more involved in their children's lives and they found the time to do so during the pandemic. Because they spent more time at home (because they did not work anymore or they worked in a hybrid system), mothers had time to do the things they weren't able to do or the things they did not realize they had to do in the period before the pandemic. Other mothers, acknowledged that physical distancing can severely impact their children, and they decided to be more involved in the relationship with them, to pay more attention to the children's needs, feelings, and to talk more with them in order to help them get over this period more easily.

Among the aspects that deteriorated during this period, we mention more frequent conflicts that they did not know how to manage, increased monitoring and control up to possessiveness which generated conflicts with children, but also the emergence of mothers' internal conflicts (mothers with at least 2 children could not manage to spend time equally with them, they neglected their relationship with their husband/partner, and thus they started to feel guilty, they find it hard to manage situations, they lacked authenticity and sometimes were not really present in the actions they were carrying out. These deteriorations appeared rather in the cases of mothers who did not have the support of the father or the support of their extended family. The quotes in S3 Appendix illustrate mothers' perceptions of these positive or negative changes in parenting behavior. In addition, we next detail the way participants experienced parenting behavior improvement or deterioration on each dimension separately.

*Bonding (effective parenting)*. Because mothers spent more time with their children, their level of empathy increased, that generated improvements in parenting behavior. This period brought a series of lessons for mothers, regarding the way they could be more effective in their parenting. Some of them learned to be more patient, they had the chance to understand better the needs of their children. Thus, in order to compensate for the negative effects of the pandemic on their children, mothers tried to do everything in their power to protect their children and to be more present, even though this required "sacrifice" and "effort", and there were times when their reactions were not "authentic".

Changes that took place at their jobs, associated with role overloading (they had to mothers but also teachers), left their mark on some mothers who found it difficult the find a balance.

During this period, some mothers lost their peace and balance, due to the fast rhythm and the density of the activities they had to carry out.

Sometimes mothers spent more time with one of their children and less with the others, and the negative emotions ("increased rage", "lack of patience, anxiety and irritability" overwhelmed them and they prevented them from being good parents. These mothers had the capacity for self-reflection and self-observation, but this puts even more pressure on them, their internal conflict reaching its peak, thus generating "crises" manifested through "guilt and feeling overwhelmed", through the idea they were not good enough, that they do not know how to be role models. Several answers highlighted the mothers' frustrations, anger, but these feelings registered the highest level in the case of a mother who wasn't living with the father of the child or who did not maintain any kind of relationship with other family members: "I feel frustrated, angry, I have the feeling I am a looser mother".

*Affection*. Some mothers felt a greater need to manifest affection towards their children, whether this need came as a result of the fear that something bad might happen to their children, or it came from the understanding of the fact that the pandemic period was difficult for children too. In other words, mothers expressed affection more frequently in order to compensate for the negative effects of isolation or physical distancing, but also because of their projections of an uncertain future. Other mothers manifested affection on purpose, without being authentic: they tried to control their negative emotions and tried to not act on them, for the emotional wellbeing of their children. Some mothers showed more affection so that children will forgive their "mistakes"- actions which mothers themselves defined as mistakes. Sometimes, too much affection led to unforeseen consequences, such as overprotection and possessiveness. The conduct of showing affection, whether it was intentional or not became a routine in the life of mothers, them realizing such positive actions should be kept in the future too.

*Communication*. The answers most mothers provided resumed at the expression: "I communicate more with the children". Those mothers who also described the ways in which communication improved, stated that they "consult with their children" with regard to certain domestic activities, certain feelings–how to express their negative emotions such as fear, anger, frustration, they watch movies together and later discuss them, they talk about COVID– 19 safety measures. Thus, during the special situations created by the pandemic, mothers managed to communicate with their children and make a common front in order to overcome the pandemic. Some mothers founded "The family council", where they discussed several topics. However, there were also cases in which mothers became overprotective due to the severity of the disease, and this conduct deteriorated the relation with their children.

On the other hand, the conflicts during this period generated by various reasons, negatively impacted the mother- child communication process, mothers often screamed and adopted a more aggressive communication style.

*Warmth/Caring*. Together with the pandemic, mothers' focus shifted on caring, and in order to face the challenges and the changes in their family life, some of them adopted a different approach, and went from an individualist approach to a common one. Hence, mothers managed to deal with the changes by developing trust, collaborating and accepting the support of the whole family, including the support of children in the case of domestic activities.

One mother stated that she did not manage to properly care for her child during the pandemic, and that she had other reasons why she decided to care more for her child. The explanation is that, due to the fact that mothers had to stay more at home, they also had to take more care of their children. Otherwise, they wouldn't have adopted such conduct. Other mothers resorted to nannies, who cooked, washed the children's clothes and took them to school, in order for them to spend more time with their children and carry out the "important" activities.

*Involvement*. Regarding the mothers' interaction with their children, the leitmotif found in the answers of mothers refers to the fact that they "tried to be present/to be there for them and to be close to them". Thus, there were several life aspects of the children that mothers were most involved: school activities, personal development and entertainment activities. However, the results of the research showed that mothers engaged most in the school activities of their children, this being the sector which mothers felt that suffered the most changes. Their children needed technical help in fulfilling tasks, some of them did not understand well the lessons and could not keep up with the online courses, and their mothers had to compensate and help children overcome these challenges. Some mothers chose to get more involved in the school life of their children because the educational process was carried out poorly, and thus they chose to complete tasks from special workbooks. There were also mothers who supported their kindergarten children with the help of online activities. In regards to the personal development of their children, it was very useful that mothers helped children with organizing their schedule, meeting dealing, they offered them support in their relationships with their friends, and they encouraged them and try to offer them a sense of freedom.

Moreover, also tried to engage in children's extracurricular activities. Most mothers used to engage in such activities before the pandemic too, but the change was in the frequency with which they did it now, namely "more often" or in the fact that "all members of the family were involved". Mothers adopted this behavior in order to protect children from spending too much time on their phones or other devices. Among the activities in which the mothers were involved are: cognitive skills development games (teaching games, board games, leggo, puzzle, reading, experiments), they played sport (volleyball, badminton), or rode bikes, did gymnastics or took long walks in nature. They also played artistic games (crafts, they built Christmas ornaments, they sang, danced, painted, watched cartoons or documentaries), and they involved children in domestic activities: cooking, washing, cutting vegetables, planting seeds). During the pandemic, some mothers had more free time and they rediscovered some of their childhood games (puzzle, team games, scrabble, chess, volleyball), and they even managed to take children to various sports activities and sometimes they invented new ones.

However, mothers also pointed out some changes with negative valences. For example, in the cases in which mothers had two or three children, they did not manage to engage in age-specific activities and usually, the youngest child was the one who was most neglected or deprived of tasks. Thus, the results of the research revealed that, the more frequent involvement in children's activities, but also the separation of types of age-specific activities where there were children of different ages led to discussions about the lack of time for the relationship with the partner. Moreover, because they were more involved in school activities, mothers who had a lot of work, had less free time for entertainment activities (just on the weekend).

Furthermore, mothers who worked exclusively online stated that due to the work related changes, they spent less time carrying out entertainment activities with their children. Even though they were spending more time at home, they used to talk very much on the phone and they were not available for their children, they were not present or authentic.

*Closeness*. Most respondents declared that during the pandemic they got closer to their children, in spite of their conflicts and frustrations. The fact that they spent more time together, offered mothers the chance to observe better their children and to see them in learning contexts that were not available before the pandemic (prior to the pandemic children were at school or at kindergarten), and to observe how emotions of their children because "the time spent online affects the children emotionally". However, some mothers re-lived the joy of the first years of childcare, when they had more time for developing entertainment activities with their children. Thus, mothers others and children became more patient with each other and they learned to cooperate and support each other. Even more, some mothers saw the pandemic

period as an opportunity to carry out certain activities that they failed to do before due to their work schedule, activities that are not related to the school sector and that made mothers be more close to their children.

*Monitoring and control.* According to the results of the research, respondents perceived this dimension in close relation with the school activities. Mothers resorted to various ways of monitoring their children, and this way varied in intensity, from "I gave them the freedom to organize themselves as they know during the online activities" and "I did not control them much", to "I vigorously monitored them because the volume of the task they had to fulfil individually was higher". Some mothers thoroughly checked the children's homework, others by surveying them or at the request of the children. Other mothers managed to impose strict rules regarding curfew, regarding homework hours, they limited the TV time of their children, or the time they were allowed to spend on their phones and tablets. Hence, such conduct implicitly determined mothers to engage more with their children, but also made them more stressed because they always had to keep an eye on the children, especially during online courses, in order to make sure they pay attention and do not play online games: "I enter in his room to check if he pays attention to the courses". Some mothers monitored their children indirectly: they heard the discussions between their children and their peers and teachers, and tactfully, they tried to talk with the children about certain aspects.

Mothers also manifested more control regarding hygiene rules: "we paid more attention to the way children washed their hands after playing outside and after touching objects". On the other hand, among the respondents there were also mothers who did not control their children at all, but they did not do it on purpose, but because the pandemic context forced them to take their children to their grandparents.

In regards to actions and behaviors which did not refer to school, most mothers did not report resorting to punishments, them considering that they did not have to punish their children because of various motives (their age, the efficiency of such methods). However, a quarter of the respondents mentioned an increase of demands and control, and stated that they did resort to punishments that increased in intensity depending on the actions of the child: from "no screen time for. . ./", "I limit his time online because otherwise he might refuse to eat or do something else", to: "we had to leave the playground and go inside after he angrily threw away another child's toys (after we repeatedly try correct his behavior), "I forbid him to play on the computer between online classes and homework", and to: "Yes, I broke his phone because I could no longer manage the time he spent in front of a screen", "Yes, unfortunately, I threw away their toys, I broke some of the toys or I hit them" (mother with high–school studies, who doesn't live with the father of the children).

Hence, because they spend more time with their children, mothers whose children were enrolled in school, started to control and monitor them more, and the children felt they had no independence at all, this conduct generated more conflicts between them.

However, mothers with children aged 10 or above consider that their children became more responsible and independent. Mother with children under 10 years old, perceived an increase in children's level of dependence on them. Knowing that they were home, the children expected their mothers to do things that were usually done by them. In addition, due to the measure of physical distancing and lack of interaction with other children, mothers also took on the role of socializing and entertaining, the role played before by the friends of their children. Thus, such behavior determined mothers to have a closer relationship with the child but it also determined children to be more dependent on their mothers.

*Conflict.* The results of the research reveal that there were many conflicts between mothers and children, even in the case of mothers who generally reported positive changes. Where an increase in the frequency of conflicts has been mentioned, the conflicts were generated either

by the time spent together, which was sometimes "too much"- a thing that made mothers control more their children, or by the children's school tasks and homework.

In the case of mothers with children enrolled in school, mothers had to take on the role of the teachers. In this regard, they became more authoritarian and this led to conflicts because children were not able to perceive them in such different roles.

Another element that generated conflicts was the mothers' lack of time due to the overload at work. Some of the mothers worked at home or in the hybrid system, but practically they worked more than before. During this period, conflicts were also generated by the dissatisfaction of their partners, which came as a result of the changes in the time and resources mothers spent on their children, but also by the fact that mothers allowed children to carry out activities they normally wouldn't let them to do, just to have time to finish their work tasks. These conflicts negatively impacted the way mothers communicated with their children: "I have never screamed so much at him as I have during this period."

Hence, the qualitative data complete the quantitative data by illustrating the fact that regardless of the general perceived trend (positive or negative) mothers have experienced both positive and negative changes. The sample of the qualitative research being highly educated, with high income, rather reflects the opinion of the mothers who were less affected by the pandemic and who had a series of resources that helped them overcome the situation. These mothers managed to get past the tense moments with the help of these resources and to mention that overall, the changes in their behavior were positive (as the quantitative data shows).

From the analysis of the mothers' responses, we identified two categories of resources: external resources (family, educational resources), or internal resources (developing relaxing coping mechanisms by talking on the phone with their friends, going out in the park, hiking or releasing negative emotions through crying, faith in God). Thus, the most important resources in this period were the ones we also identified through the quantitative data: the partner/father of the child, and the grandparents. The father and the grandparents offered mothers support by taking responsibility for various activities of caring for and educating children, but they also offered them emotional and informational support. In families with more than one child, the oldest child (regardless of his/her age), became an important resource, because he/she used to play with the younger children, and had to watch over them long enough for mothers to fulfil other tasks. Even more, children were considered important resources when mothers felt overwhelmed, and the mothers were recharging with the "children's energy and enthusiasm". In order to find strategies for managing internal conflicts or conflicts with their children, mothers also resorted to specialized educational resources (books/articles, they listened to podcasts and interviews with a specialist, they took online courses or resorted to supporting groups, leisure activities or even to psychologists / psychotherapists).

## Conclusions and discussions

The purpose of our paper was to analyze the changes in the parenting behavior of Romanian mothers whose children are under the age of 18 in the context of the pandemic. On the basis of the theoretical framework, we assumed that by modifying some external variable, mothers' level of stress will increase and this will lead to negative changes in their parenting behavior. The results of the research revealed that only mothers who experienced a negative impact of the pandemic changed their behavior in a negative manner, especially those mothers who did not have social support or parenting alliance, who had a lower level of education and low, below average incomes. These results are in line with other studies which show that these types of mothers, due to their struggle to balance their caregiving duties with their professional

duties tend to adopt negative behavior towards their children [45] and with studies which show that parents who were more stressed, tended to spend less time with their children, to no longer offer them the attention needed, and to no longer take into account the emotional well-being of their children [46].

Moreover, previous studies found that in general the parents with lower incomes and education registered more parenting stress, more engagement in negative behaviors, which manifested itself during the pandemic as well [47–52]. The mothers who were more affected by the pandemic perceived a deterioration of their parenting behavior and resorted to disciplinary practices more frequently. In this regard, the qualitative data of our study illustrates the experiences of mothers. Anger and frustration are the leitmotifs of mothers, and such emotions sometimes determined mothers to be more authoritarian, to increase the frequency of the disciplinary practices,which sometimes led to faulty communication (screaming, aggressively) to limiting certain activities (time spent on digital devices, or times spent watching TV). Sometimes these feelings made mothers overprotective and they led to excessive involvement in children's school life, but they also made some mothers neglect their children or offer them too much freedom. Such results were also found in previous studies [46, 50, 53].

In the context of resources, mothers felt overwhelmed, them reporting that they had too many responsibilities compared to their resources of time and energy [54]. Hence, the social support from other family members (grandparents) or support of the father of the children, decreased mothers' levels of anxiety and in this way their behavior was less affected or deteriorated, results also revealed by other studies [47, 48]. Usually social support serves as a buffer between life events and psychological distress [55, 56], and even more in the pandemic context [57], although social support has been reduced only to two categories of people previously mentioned. The qualitative data supports these results, by showing that in the cases of mothers who did not have social support, especially from the fathers of the children or their grandparents, they encountered more difficulties during the time of the pandemic [58, 59].

Mothers with primary education and low incomes have had a harder time coping with the pandemic, and this has resulted in some deterioration in parenting behavior and an increase in discipline practices. Thus, these mothers were affected by the pandemic, and as previous studies show, in such situations, their disciplinary practices increase [60, 61]. In the case of mothers who had proper access to external resources, the impact of the pandemic was significantly lower, and such results are in line with the results of other studies [59, 62, 63].

Previous studies revealed that mothers' perception of the impact of the pandemic on their lives, can be influenced by various elements including income, living conditions, the level of perceived social support or social cognitive capacities [64]. Hence, mothers with higher levels of education, with higher incomes, who implicitly have better developed parental skills, perceived the pandemic in a more positive way, as an opportunity to spend more time with children [58], and to develop their parenting behavior. Our research also supports these results, by showing that mothers with high levels of education, and medium- high income levels, perceived the changes generated by the pandemic as an opportunity to foster and develop a better relationship with their children, them also reporting rather positive changes in their parenting behavior.

Most mothers had a series of resources that helped them develop coping strategies, especially mothers with higher education studies, who were married or had a partner [65]. Thus, the qualitative data revealed that these mothers were willing to find other external resources than the one found in the quantitative research. In this regard, it is necessary to mention that the language used by the respondents, "betrays" and shows not only that they resorted to psychologists/psychotherapist, but also to specialized literature on parenting, emotional and crisis communication, psychological development. Even more, the self–didactic approach is

"betrayed" by the clumsiness in using specialized terms. In this regard, the results can suggest that Romanian mothers (at least those with higher education studies), try to move beyond the traditional approach to parenting, and parent—child relationships, and try to have a more modern approach that also has a scientific basis. We can relate this change of perspective, to the pandemic, by taking into account the fact that mothers declared they read more, but we have to take into account the fact that they did not mention what books they read. Hence, these mothers were willing to seek solutions and information, and such conduct helped them find ways to deal with the challenges generated by the pandemic, and grow and foster personal development. In this regard, mothers were more confident, and similar to previous studies, confidence is very important because it has an essential role in helping mothers to prevent the emotional disruptions of their children [66]. Taking into account the aspects previously mentioned, we can infer that mothers with higher education studies and higher incomes perceived an improvement in their behavior in relation to their children, and the results of our research are in line with previous studies [48, 58, 67].

Mothers who fall within the category of mothers with higher educational levels and higher income, became more close to their children, and they felt more gratitude towards them, and they pampered and offered them more support. Sometimes, they spent quality time with their children (they carried out leisure activities, personal development/cognitive development activities), but sometimes the time spent with children was not qualitative: mothers were not authentic, and were not really available and willing to carry out certain activities with them.

Most mothers communicated more with their children and adopted an assertive communication. They manifested their affection more often in order to offer emotional support for their children. Thus, they were more patient and they tried to manage their negative emotion in order to keep the family's harmony and peace. Even more, mothers resorted to disciplinary practices less or the same as prior to the pandemic period, and this result is in line with previous studies [67–70].

Although the respondents' tendency was to emphasize the positive changes, mothers also highlight some aspects of their relationship with their children which deteriorated. The qualitative research reveals that mothers, including those who generally reported positive changes in their behavior, experienced negative changes, this result being also reported in other studies [71].

Similar to previous research [72], our research shows that regardless of the mothers' socio-demographic characteristics, there was greater involvement in school life, especially for mothers with children under 10 years old. Moreover, in the context of school closure, previous studies also reveal that parents were required to offer more attention to the education of their children: they had to aid them in doing their homework or completing tasks, because in the online environment children tended to be more resilient to asking questions during classes, about the aspects they did not understand [73]. In this regard, the increased level of monitoring and involvement of mothers in the school activities of their children generate conflicts, especially conflicts related to organizing their schedule or fulfilling tasks. One of the elements that led to conflicts was also the long time children spent on digital devices, result which was revealed by other studies too [49, 50]. Furthermore, similar to previous research, [74] the qualitative data show an increase of conflicts between parents and their children, during the pandemic. Other studies did not reveal increased conflicts compared to the time before the pandemic, but such studies were conducted in the early pandemic period [68].

Therefore, our paper also has theoretical and practical implications. From the perspective of the theoretical implications, our paper contributes to the literature on changes in parent–child relationships and parenting behavior during the pandemic, by providing insights on the way a crisis situation like the pandemic influenced the parenting behavior of mothers in

positive and negative manners. Even more, considering the Romanian context, since very few studies approached the subject of parenting behavior, our paper could also fill in the gap which refers to how mothers behaved towards their children in the pandemic period. In regard to the practical implications, by emphasizing both the positive and negative changes in the behavior of parents in the pandemic period, the paper can be used as a frame of reference for future research, and it also draws attention to the challenges that mothers have faced in this time of crisis, and promotes some of their positive practices, which could also be adopted by other mothers.

## Limitations and future research directions

The results and implications of our study should be taken into consideration by bearing in mind the limitations of the research. Firstly, one limitation is represented by the fact that the sample of convenience was predominantly composed of well educated, employed mothers, with medium to high income, who were interested in the topic approached. In this regard, the results rather reflect the perception of the mothers who pay attention to their personal development and who resort to specialized books or professionals in the field. Future research could focus more on other types of mothers, and could also analyze the parenting behavior of fathers during the pandemic.

Secondly, in our analysis, we did not take into account a number of variables that could have impacted the dynamics of the mother-child relationship such as: the quality of family relationships, the age of the mother or the characteristics of children (if they have certain conditions, disabilities, different temperaments). Future research could exceed these limits by including these aspects in the analysis.

Another limitation of the study is represented by the fact that we did not have a research conducted prior to the pandemic to which to report. In the absence of this zero or control point, the analysis was performed based on mothers' subjective perceptions of their changes in parenting behavior, and the classification process of positive and negative changes was conducted only by considering the impact of these changes on the children. Even more, mothers perception of how they should act during the pandemic was influenced by the information read or hear on mass–media channels. In this regard, it is questionable to what extent certain behavioral changes have led to impairment of physical and mental health and what long-term repercussions these changes have had on mothers, and we considered that this could be another future research direction.

## Supporting information

**S1 Appendix. Descriptive statistics of the index's items.**
(DOCX)

**S2 Appendix. Sociodemographic characteristic of qualitative survey respondents (N = 24).**
(DOC)

**S3 Appendix. Results of the qualitative thematic analysis.**
(DOCX)

## Acknowledgments

We kindly thank the respondents for participating in the study.

## Author Contributions

**Conceptualization:** Luiza Mesesan-Schmitz, Carmen Stanciu.

**Data curation:** Luiza Mesesan-Schmitz.

**Formal analysis:** Luiza Mesesan-Schmitz, Carmen Stanciu, Venera Bucur.

**Investigation:** Carmen Stanciu, Venera Bucur, Laurentiu Gabriel Tiru.

**Methodology:** Luiza Mesesan-Schmitz.

**Project administration:** Luiza Mesesan-Schmitz.

**Resources:** Luiza Mesesan-Schmitz, Maria Cristina Bularca.

**Supervision:** Claudiu Coman.

**Writing – original draft:** Luiza Mesesan-Schmitz, Maria Cristina Bularca.

**Writing – review & editing:** Luiza Mesesan-Schmitz, Claudiu Coman, Carmen Stanciu, Venera Bucur, Laurentiu Gabriel Tiru, Maria Cristina Bularca.

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
