## [Decision Letter · Decision Letter 0]

17 Oct 2023

PONE-D-23-19603Changes in Parenting Behavior in the time of COVID – 19: a mixed method approachPLOS ONE

Dear Dr. Coman,

Thank you for submitting your manuscript to PLOS ONE. After careful consideration, we feel that it has merit but does not fully meet PLOS ONE’s publication criteria as it currently stands. Therefore, we invite you to submit a revised version of the manuscript that addresses the points raised during the review process.

Please note that we have only been able to secure a single reviewer to assess your manuscript. We are issuing a decision on your manuscript at this point to prevent further delays in the evaluation of your manuscript. Please be aware that the editor who handles your revised manuscript might find it necessary to invite additional reviewers to assess this work once the revised manuscript is submitted. However, we will aim to proceed on the basis of this single review if possible. 

We look forward to receiving your revised manuscript.

Kind regards,

Avanti Dey, PhD

Senior Staff Editor

PLOS ONE

Reviewers' comments:

Reviewer's Responses to Questions

**Comments to the Author**

1. Is the manuscript technically sound, and do the data support the conclusions?

Reviewer #1: No

2. Has the statistical analysis been performed appropriately and rigorously? 

Reviewer #1: No

3. Have the authors made all data underlying the findings in their manuscript fully available?

Reviewer #1: Yes

4. Is the manuscript presented in an intelligible fashion and written in standard English?

Reviewer #1: No

5. Review Comments to the Author

Reviewer #1: This study examines changes in parenting behavior due to COVID-19 among a sample of parents in Romania. While the topic of this research is needed and compelling, the setup for the study – from the literature review to the study design, to the methodological approach – is insufficient and not rigorous enough to substantially contribute to the literature. My specific comments and recommendations are below.

Abstract

- Please add more details about the Romanian context, and why research on parenting changes in Romania is needed.

Introduction / Literature Review

- Please provide more detail on the Romanian context in the middle of COVID, with special attention to school policies, return-to-work timelines, and country-level precautions (and how these policies relate to parental adjustment)

- Please provide more detail about the parenting stress model – for example, providing illustrative examples; more accurately describing the environmental/behavioral/developmental variables the authors mention. The authors should specifically focus on the environmental, behavioral, and developmental measures that they included in their model.

- Similar to above, the authors need to go into more details of defining and describing “social support, parenting alliance, parenting skills, material resources, and cognitive coping.”

- I personally find the “The concept of parenting behavior” disorganized and lacking an overall framework. The authors begin by discussing parenting styles/dimensions, then go into parental support and control, then positive and negative parenting, then positive parenting. This section needs to be edited to have a better focus and a better argument for why the authors are focusing on the specific parenting behaviors they have chosen in their study.

- Related to above, the literature review on disciplinary practices is insufficient. The authors need to discuss the specific disciplinary practices that have been shown to positively or negatively impact child outcomes.

- The authors need to provide a more convincing argument of the gaps that they are filling in the literature by conducting this study. I find the literature review largely disconnected from the vast amount of research that has already been conducted on parenting behavior during COVID-19.

- Even though “parent involvement in school” is a key dependent variable in the author’s analysis, there’s virtually no mention of this in the introduction or lit review. The authors need to formulate an argument to discuss why measuring parental involvement in school is needed (and then connect that to prior literature that has looked at this variable).

Methods

- As written, H4 is not framed as a hypothesis.

- As written, in the introduction, the authors use the parenting stress model and describe how environmental resources direct influence parenting behavior. However, in H4, the authors introduce a moderating relationship. The authors would need to provide justification in the setup of the manuscript for a moderation analysis.

- I’m confused why the authors have two objective, four hypotheses, but only one research question.

- Please provide more detail about the “varied” facebook groups.

- I feel the authors should provide more justification for the way they collected their data. There are strengths and weaknesses to using social media groups, and these should be discussed – ideally, this should be introduced in the introduction/literature review section, and they authors would set up an argument for how collecting this type of data responds to gaps in the current literature.

- It looks like the authors made a parenting behavior index themselves? I’m confused why this was chosen, given there are many cross-culturally validating parenting indices to use. It’s now difficult to assess the validity and usefulness of these results, given that this scale has not underwent a rigorous scale validation process (a factor analysis is and chronbach’s alpha is insufficient).

- Please list out the 7 disciplinary practices that were measured (again, these should also be introduced in the introduction).

- It is unclear why the authors chose to measure specific resources they did in the study (i.e., social support, parenting alliance, parenting skills competencies, cognitive coping). There would need to be a much clearer argument from the introduction before these measures are introduced in the Methods section.

- It appears that the qualitative coding was not conducted in a sufficiently rigorous fashion; there should be at least 2 independent coders, and inter-coder agreement should be calculated.

Results

- In my opinion, a correlation analysis is an insufficient method with which to answer the research question. This should at least be a multivariate regression, with a number of control variables included, as informed by the literature.

- Again, a sub-group analysis where the authors run a simple correlation is not a sufficiently rigorous way to answer this research question. The authors should have conducted a multivariate regression, and then introduced interaction terms (e.g.,, CovidImpact * SocialSupport), to determine if the relationship between COVID impact and parenting behaviors vary by social support.

6. PLOS authors have the option to publish the peer review history of their article (what does this mean?). If published, this will include your full peer review and any attached files.

Reviewer #1: No

---

## [Author Response · Author response to Decision Letter 0]

12 Dec 2023

Rebuttal letter

 Claudiu Coman

Transilvania University of Brasov

claudiu.coman@unitbv.ro

Dear Sir/Madam

With this cover letter we submit the revised manuscript, entitled “Changes in Parenting Behavior in the time of COVID – 19: a mixed method approach”, by Luiza Mesesan-Schmitz, Claudiu Coman, Carmen Stanciu, Venera Bucur, Laurentiu Gabriel Tiru, and Maria Cristina Bularca for publication in PLOS ONE.

We revised the manuscript according to the suggestions and recommendation made by the reviewer. We would like to thank the reviewer for taking time to review our paper and for providing such useful suggestions. We also thank the academic editor for reviewing our paper. We tried to comply with all the suggestions and recommendations made by the reviewer, and in this letter, we describe the changes we made to the text according to the recommendations of the reviewer. The changes we made to the text were made while having active the “Track changes” function from Microsoft Word and the text was changed can be best seen while having active the “All markup” option. 

Response to Reviewer 1 comments

Reviewer 1 comment:

This study examines changes in parenting behavior due to COVID-19 among a sample of parents in Romania. While the topic of this research is needed and compelling, the setup for the study – from the literature review to the study design, to the methodological approach – is insufficient and not rigorous enough to substantially contribute to the literature. My specific comments and recommendations are below.

Response: 

We firstly thank the reviewer for taking time to review our manuscript again and provide suggestions in order to improve it. We addressed all the suggestions made by the reviewer. The changes we made to the text can be best seen while having active the “ Track changes” function and the “All markup” option. 

Reviewer 1 point 1: Abstract - Please add more details about the Romanian context, and why research on parenting changes in Romania is needed

Response 1: We are thankful to the reviewer for the suggestion. In order to comply with the suggestion, within the Abstract of the paper we added information about the Romanian context and shy research on parenting changes was needed at that time in Romania. 

Thus, at page 1 of the manuscript (with “Track changes” and “All markup” active), in the Abstract, we added the following text:

“Research on parenting changes was important in the Romanian context because, in that challenging period, there were no regulations to safeguard parents, especially single parents as mothers. Mothers experienced increased levels of stress, some of them having to leave their jobs to stay at home with their children. Other mothers needed to work from home and in the meantime to take care of their children. In this context we wanted to illustrate the possible changes that occurred in their parenting behavior during the pandemic period.” 

Reviewer 1 point 2: Introduction / Literature Review - Please provide more detail on the Romanian context in the middle of COVID, with special attention to school policies, return-to-work timelines, and country-level precautions (and how these policies relate to parental adjustment)

Response 2: We are grateful to the reviewer for the comment. We would like to mention that during the pandemic period, in Romania there were no clear school or return to work policies. Decisions were made in short time, and each school/university had to manage the situation on its own manner. However, in order to comply with the suggestion of the reviewer we provided the description of some measures imposed by the Romanian Government in order to prevent and combat the effects of the COVID-19 pandemic within educational institutions.

In this regard, at pages 3-4 of the manuscript (with “Track changes” and “All markup” active), in the Introduction section of our manuscript, we inserted the following text: 

“Thus, in the context of school or return to work policies, according to Law nr.55 from 2020 of the Romanian Parliament for preventing and combating the effects of the COVID-19 [2], within educational institutions the measures which were imposed were general, they depended on the evolution of the pandemic and on each educational institution. One of the measures stated that: subject to the analysis of the epidemiological situation at national level carried out by the Ministry of Health and based on the decision of the National Committee for Emergency Situations, by order of the Minister of Education, the suspension of activities that require the physical presence of pre-school children, preschoolers and students in educational units can be ordered, and the didactic activities will continue in the online system. Other measures stated that: during the state of alert, which was still available in 2021, and until the restrictions on public gatherings are removed by the relevant authorities, the pre-university education units organize activities from the education plans in the online environment, that in order to ensure equal access to education, school inspectorates and pre-university education units have the obligation to ensure educational resources for students who do not have access to technology, in accordance with the instructions of the Minister of Education and Research, and that the national exams of students, which involved face to face interaction, had to be taken face to face but under health protection conditions [2].”

Furthermore, because we added information to our text from a new source, we inserted the source in the text and in the References section. In this regard, all the numbers of the references within the text changed, and the changes can be seen while having active “Track changes” function and “All markup” option from Microsoft Word.

The reference we added is the following: 

[2]. Romanian Parliament, LAW no. 55 of May 15, 2020 regarding some measures to prevent and combat the effects of the COVID-19 pandemic. [cited 2023 Nov 20] Available from: https://legislatie.just.ro/Public/DetaliiDocumentAfis/245554

Reviewer 1 point 3: Please provide more detail about the parenting stress model – for example, providing illustrative examples; more accurately describing the environmental/behavioral/developmental variables the authors mention. The authors should specifically focus on the environmental, behavioral, and developmental measures that they included in their model.

Response 3: We are very grateful to the reviewer for the suggestion. However, we believe we have described the variables/measure on which we focused our study, in the context of the Parenting Stress Model developed by Abidin. [11. Abidin RR. The Determinants of Parenting Behavior. Journal of Clinical Child Psychology. 1992;21(4):407–412 doi: 10.1207/s15374424jccp2104_12]. Thus, at page 7 of the manuscript (with “Track changes” and “All markup” active), in section: Theoretical framework, we described the model developed by Abidin, stating that “While recognizing the influence of environmental, behavioral or developmental variables, the author also mentions that the parenting behavior is mediated by a series of resources.” Then, we described the parenting role variable, and the parenting stress. According to the model, parenting role is influenced by elements such as work, marital status, daily hassles or life events, and the parenting stress variable is influenced by elements such as: social support, parenting alliance, material resources. Through the text mentioned above, which can be found in our manuscript at page 7, we tried to describe and show on which variables we focused in our study (the elements from the parenting role and the parenting stress variables, which together form the parenting behavior variable). We thank again the reviewer for the suggestion.

Reviewer 1 point 4: Similar to above, the authors need to go into more details of defining and describing “social support, parenting alliance, parenting skills, material resources, and cognitive coping.”

Response 4: We thank the reviewer for the suggestion. However, according to the model we used [11. Abidin RR. The Determinants of Parenting Behavior. Journal of Clinical Child Psychology. 1992;21(4):407–412 doi: 10.1207/s15374424jccp2104_12]., these variables: “social support, parenting alliance, parenting skills, material resources, and cognitive coping”, are resource variables, and the way they were measured is described in our manuscript. In this regard, the variables and how we measured them are presented in section Materials and Methods, subsection Measures, pages 17-22 of our revised manuscript with “Track changes” and “All markup” active. 

Reviewer 1 point 5: Similar I personally find the “The concept of parenting behavior” disorganized and lacking an overall framework. The authors begin by discussing parenting styles/dimensions, then go into parental support and control, then positive and negative parenting, then positive parenting. This section needs to be edited to have a better focus and a better argument for why the authors are focusing on the specific parenting behaviors they have chosen in their study.

Response 5: We are very grateful to the reviewer for the suggestion. However, in the section which refers to Parenting Behavior, we firstly described the parenting styles, then the dimensions of the Parenting Behavior, and we found in the literature a consensus on two main dimensions: parental support and parental control. Then, we offered details about these two dimensions. Furthermore, we presented the positive and negative aspects of the parenting behavior, by describing and giving examples of positive and negative parental practices. Then, we mentioned that the positive/negative behavior of parents can be found in the literature under the concept of positive/poor parenting. In this regard, we defined positive parenting and poor parenting. Thus, we wanted to provide a comparative overview of the positive and negative practices of parents, and this is why the text was organized in this way within the subsection. Therefore, in our study we focused on the main dimensions of the parenting behavior, concept found in the literature and the focus was on the dimensions of parenting behavior which we considered could undergo changes in the pandemic period. 

Reviewer 1 point 6: Related to above, the literature review on disciplinary practices is insufficient. The authors need to discuss the specific disciplinary practices that have been shown to positively or negatively impact child outcomes.

Response 6: We thank the reviewer for the suggestion. In order to comply with the suggestion of the reviewer we looked for other studies that defined disciplinary practices and we inserted information from them in our manuscript.

In this regard, at pages 10-11 of the manuscript (with “Track changes” and “All markup” active), section: The concept of parenting behavior, we inserted the following text:

“Considering the discipline practices, they are methods through which parents try to control the children's negative behavior and promote their positive behavior. In this regard, these practices could be violent or non – violent [25] and some of the discipline practices include: teaching children about good and bad behavior, making the child apologize, giving children time –out, taking away their privileges, using corporal punishment, expressing disappointment, shaming, yelling, withdrawing love for misbehavior, threatening them or promising a treat or a privilege [26]. Given the impact of such practices on children, a previous study conducted on mothers and children[26], found that practices such as corporal punishment, expressing disappointment and yelling were associated with more child aggression, and also that corporal punishment, expressing disappointment and shaming were associated with more child anxiety. Another study [25], conducted by UNICEF on mothers and children from low and middle – income countries, revealed that non-violent practices, mainly explaining why a behavior is wrong, were generally the most common practices used by mothers or caregivers, and only one in four mothers/caregivers believed that physical punishment (such as shaking the child, slapping the child) is needed in order to educate and manage the behavior of their children. Thus, taking into account the two categories of disciplinary practices, in our research we paid attention to the non-violent disciplinary practices, which include acts such as taking away privileges or explaining why something is wrong. Alt these kind of acts can have a positive impact on the outcomes of the children, and we supposed that during the pandemic, any changes in the frequency of these acts can increase frustration and negative emotions of the children and can have a negative impact on them. Details about the disciplinary practices that were measured in our study can be found in Appendix S1.”

In this regard, we added to our study information from two other sources, which were cited in the text and in the References section. 

The references included in the revised manuscript are:

[25]. UNICEF, Child Disciplinary Practices at Home: Evidence from a Range of Low- and Middle-Income Countries [cited 2023 Nov 20] Available from: https://data.unicef.org/resources/child-disciplinary-practices-at-home-evidence-from-a-range-of-low-and-middle-income-countries/

[26]. Gershoff ET, Grogan-Kaylor A, Lansford JE, Chang L, Zelli A, Deater-Deckard K, Dodge KA. Parent Discipline Practices in an International Sample: Associations With Child Behaviors and Moderation by Perceived Normativeness. Child Development, 2010; 81(2):487–502 doi:10.1111/j.1467-8624.2009.01409.x

Reviewer 1 point 7: The authors need to provide a more convincing argument of the gaps that they are filling in the literature by conducting this study. I find the literature review largely disconnected from the vast amount of research that has already been conducted on parenting behavior during COVID-19.

Response 7: We are grateful to the reviewer for the suggestion. In regards to the gaps that our paper is filling in the literature, we would kindly like to mention that at the time we firstly submitted our paper to this journal, there very few articles which approached the subject of parenting behavior during the COVID – 19 pandemic. In the meantime, more articles were published on this subject, but even so, in the Romanian context, there aren’t many studies which discussed this subject. We inserted in our paper a description of the studies we found on this matter, which were conducted in the Romanian context. The studies [11,12,13], mostly focused on the negative feelings of parents, on the relationship between technology and parenting styles and parent- child relationships. Therefore, our study revealed the changes that occurred in the parenting behavior of mothers during the pandemic, and the possible variables which influenced these changes. We also improved the explanation we had in our manuscript, regarding the theoretical implications of our paper, which was written at page 39 of our manuscript (with “Track changes” and “All markup” active).

Thus, at page 6 of the revised manuscript (with “Track changes” and “All markup” active), section: Introduction, we inserted the following text:

“Considering the parenting behavior in the Romanian context during the pandemic, very few studies were conducted on this subject. A previous study conducted on parents and children from the epicenter of the COVID -19 outbreak in Romania- Suceava [11], showed that,compared to the period prior to the pandemic, parents and children focused more on pending time in free time activities, chores, or social interactions, that they encountered difficulties in maintaining social dynamics, managing emotions, dealing with school issues, or motivating children to comply with parental norms. Another study, which focused on the feelings and personality traits of parents of primary school pupils [12], revealed that, during the pandemic, as the parents’ level of anxiety increased, their anger level also increased, but their level of self-efficacy decreased. Another study, which focused on the parent – child relationship and their use of technology and media [13], discov

---

## [Decision Letter · Decision Letter 1]

28 Mar 2024

Changes in Parenting Behavior in the time of COVID – 19: a mixed method approach

PONE-D-23-19603R1

Dear Dr. Coman,

We’re pleased to inform you that your manuscript has been judged scientifically suitable for publication and will be formally accepted for publication once it meets all outstanding technical requirements.

Kind regards,

Giulia Ballarotto

Academic Editor

PLOS ONE

Additional Editor Comments (optional):

Reviewers' comments:

Reviewer's Responses to Questions

**Comments to the Author**

1. If the authors have adequately addressed your comments raised in a previous round of review and you feel that this manuscript is now acceptable for publication, you may indicate that here to bypass the “Comments to the Author” section, enter your conflict of interest statement in the “Confidential to Editor” section, and submit your "Accept" recommendation.

Reviewer #2: (No Response)

2. Is the manuscript technically sound, and do the data support the conclusions?

Reviewer #2: (No Response)

3. Has the statistical analysis been performed appropriately and rigorously? 

Reviewer #2: (No Response)

4. Have the authors made all data underlying the findings in their manuscript fully available?

Reviewer #2: (No Response)

5. Is the manuscript presented in an intelligible fashion and written in standard English?

Reviewer #2: (No Response)

6. Review Comments to the Author

Reviewer #2: Thank you for the opportunity to review this interesting study. The authors responded well to reviewer 1's suggestions, significantly improving the paper. I think the study is interesting and can be published

7. PLOS authors have the option to publish the peer review history of their article (what does this mean?). If published, this will include your full peer review and any attached files.

Reviewer #2: No
